# Ultimate Context of the Termination of Parental Investment

**DOI:** 10.3390/ijerph22060944

**Published:** 2025-06-16

**Authors:** Josip Hrgović

**Affiliations:** Department of Public Relations, University North, Dr. Žarko Dolinar Sq., 48000 Koprivnica, Croatia; josip.hrgovic@unin.hr

**Keywords:** parental investment, socioeconomic status, child mortality, human behavioral ecology

## Abstract

This paper investigates the ultimate socioeconomic causes underlying the termination of parental investment in humans by analyzing the relationship between socioeconomic status and various forms of child mortality, including live births, stillbirths, infant deaths, and infanticide. Utilizing theoretical foundations from human behavioral ecology, the study illustrates how different forms of termination of parental investment can be viewed as points along a continuum of adaptive strategies aimed at optimizing reproductive fitness. The research emphasizes that technical and cognitive limitations lead to many instances of infanticide being concealed as natural child deaths, such as Sudden Infant Death Syndrome (SIDS), thus complicating the accurate detection of true causes of death. However, addressing common ultimate causes—specifically socioeconomic factors such as healthcare accessibility, nutritional quality, social support, and stress reduction—can simultaneously prevent or reduce all forms of investment termination. The paper further analyzes demographic data from Zagreb and surrounding municipalities. Ultimately, the study advocates a holistic approach to public health interventions and policies aimed at improving socioeconomic conditions as a crucial step toward reducing all forms of child mortality.

## 1. Introduction

The cessation of parental investment has always evoked strong emotions and moral judgments, yet evolutionary studies and human behavioral ecology demonstrate that this phenomenon is universally present across history and various cultures [1,2]. This paper assumes that the same ultimate causes, primarily socioeconomic in nature, underlie the observed patterns in live birth rates, stillbirth rates, infant mortality, and infanticide. Although these phenomena might appear distinct and specific at the proximate level, they actually represent different points along a continuum of parental investment, ranging from complete commitment to extreme withdrawal of investment [3].

Due to technical and cognitive limitations, numerous cases of infanticide and child deaths are often categorized as natural deaths, such as Sudden Infant Death Syndrome (SIDS) or fatal neglect, leaving actual incidence rates unknown and potentially underestimated [4,5]. Nevertheless, despite the inability to precisely detect every individual case, it is clear that socioeconomic factors such as limited access to healthcare, poor nutrition, elevated stress levels, lack of social support, and poverty create conditions conducive to all these outcomes [6,7,8].

The theoretical framework of this study integrates human behavioral ecology with demography, forming what is termed evolutionary human demography. Employing the phenotypic gambit [9], this approach emphasizes the importance of examining ultimate causes, specifically evolutionary adaptations, while minimizing the focus on immediate (proximate) mechanisms. Ultimate causes, referring to adaptive strategies aimed at optimizing reproductive fitness, elucidate why parents adjust their behavior according to environmental conditions—often through radical measures such as infanticide or neglect [10,11].

Throughout history and across diverse cultures, parental investment cessation has elicited profound emotional responses and moral judgments. Historical examples, such as the widespread abandonment of infants during the European Middle Ages due to extreme poverty or the selective infanticide practices documented among the Inuit communities, illustrate vividly how severe socioeconomic pressures have consistently shaped reproductive decisions. This paper explicitly assumes, grounded in robust evolutionary theory and demographic evidence, that similar ultimate causes—primarily socioeconomic factors—underpin the varied manifestations of parental investment termination observed in live births, stillbirths, infant mortality, and infanticide.

This paper argues that by addressing socioeconomic factors, which are ultimate causes, we can simultaneously prevent and reduce the frequency of all forms of parental investment cessation, even when proximate causes for individual cases remain unclear or technically inaccessible. Through this approach, it becomes possible to better understand and effectively reduce all forms of child mortality, thereby improving overall population dynamics and societal health.

The key contribution of this review lies in systematically applying a human behavioral ecology framework to unify a range of adverse reproductive outcomes—stillbirths, infant mortality, and infanticide—under the continuum hypothesis. This integrative perspective allows for more coherent theoretical explanations and potentially more targeted public health interventions.

## 2. Two Causes by Mayr and Four Questions by Tinbergen

Evolutionary approaches to studying human behavior are interpreted through the dichotomy of proximate and ultimate causes. Ernst Mayr [12] introduced a critical distinction in explaining biological phenomena: proximate (immediate) and ultimate (evolutionary) causes. His theory provided a foundation for understanding complex behavioral and physiological processes in organisms from two complementary perspectives—how and why something exists and functions. Proximate causes refer to the immediate mechanisms determining an organism’s behavior or physiological function. They explain “how” a particular behavior occurs within an individual, including the biological, developmental, and situational factors involved.

Ultimate causes, on the other hand, refer to evolutionary reasons behind the development and retention of certain behaviors across generations. They involve adaptations that enhance reproductive success. An ultimate explanation addresses “why” a behavior evolved—what selective advantage it provided.

Using infanticide as an example, proximate causes (e.g., postpartum depression, impaired mental health, cultural stigmatization of a child’s sex) clarify the immediate mechanisms leading to such acts. However, it is crucial to recognize that, at a deeper level, the same ultimate evolutionary pressures—related to limited resources and the survival of parents (as gene carriers)—ultimately drive decisions to cease investing in offspring. When parents perceive their reproductive or personal futures as endangered due to severe ecological, economic, or social circumstances, investment in offspring may drastically decline. This cessation of investment can manifest as direct child death (infanticide, starvation, neglect) or as institutionalization and similar actions.

Nikolaas Tinbergen expanded upon Mayr’s binary distinction [13]. He proposed that every biological behavior should be analyzed through four distinct yet interconnected levels of explanation. Known as Tinbergen’s Four Questions, this framework facilitates a comprehensive understanding of behavioral and physiological phenomena:

Mechanism (causation—“how does it function?”): This question examines the immediate causes of behavior—physiological and chemical mechanisms enabling the response.

Ontogeny (development—“how does it develop?”): The ontogenetic question addresses how behavior develops within an individual—from conception through adulthood.

Function (adaptive value—“why is it beneficial?”): The functional question emphasizes the adaptive purpose of behavior—how it contributes to survival and/or reproduction, focusing on adaptation and natural selection.

Phylogeny (evolutionary history—“how did it evolve?”): The phylogenetic question explores the evolutionary history of behavior: when it first appeared, how it changed over time, and its relationship to behaviors in related species.

The first level of explanation, mechanism, refers to the physiological and biological processes that enable specific behaviors [14]. In humans, mechanisms such as hormonal changes or stress responses can directly shape parental decisions and behaviors regarding investment or disinvestment in offspring. For example, research indicates that elevated stress levels and altered hormonal regulation can lead to reduced parental investment or an increased propensity for neglecting offspring [15].

The second level, ontogeny, focuses on how behaviors develop throughout an individual’s life cycle [14]. In the context of parental investment, a developmental perspective allows the analysis of how early life conditions, such as socioeconomic circumstances or the health status of a child, influence parental investment strategies. For instance, parents who experienced poverty or unstable conditions during their own childhood may later exhibit different patterns of investment or disinvestment in their offspring [16].

The third level, function, addresses the locally adaptive purpose of behavior, including the assessment of costs and benefits regarding reproductive fitness [14]. In terms of parental disinvestment, this level explains how parents make decisions based on an implicit evaluation of the expected benefits for offspring survival and their future reproductive prospects. For example, parents might reduce investment in a child with lower survival chances to reallocate resources toward offspring with higher reproductive potential, thus maximizing their overall fitness [17].

The fourth level of explanation, phylogeny, investigates the evolutionary history of behavior through the evolutionary past of the species [14]. Parental disinvestment can be studied by comparing it with similar behaviors in other species and understanding the evolutionary context of these behaviors. For example, a phylogenetic perspective helps explain why certain forms of parental disinvestment occur in humans by comparing them with similar behavioral patterns in other mammals and primates, highlighting common evolutionary selective pressures or specific adaptive responses [18].

Integrating these four Tinbergen levels of explanation allows a more comprehensive understanding of how various forms of parental disinvestment, although radically different at a proximal level, share a common ultimate logic of reproductive fitness optimization. Specifically, even though individual cases of disinvestment (such as infanticide or neglect) may arise from different immediate causes, the shared ultimate goals of these decisions always relate to adaptation to specific ecological, socioeconomic, and biological conditions that enhance parental reproductive success [14].

Instances of parental disinvestment are not random or pathological phenomena but complex adaptive responses of parents to their specific life circumstances. This integrative approach facilitates a more precise identification of conditions under which certain forms of disinvestment emerge, contributing to the understanding of variability in human parental behavior within and between populations. Simultaneously, this framework provides a clear distinction between immediate (proximate) and evolutionarily rooted (ultimate) causes of behavior, enabling a detailed analysis of the complexity of human demographic strategies.

Different forms of parental disinvestment, despite apparent differences in their immediate manifestations, reflect a common adaptive logic of evolutionary fitness optimization. This contributes to a deeper understanding of demographic phenomena associated with parental investment, bridging various disciplinary perspectives such as evolutionary biology, anthropology, sociology, and demography.

## 3. Ecology of the Termination of Parental Investment

The theory of parental investment and sexual selection, developed by Robert Trivers [10], provides a robust theoretical framework for understanding diverse reproductive strategies among sexually reproducing species, including humans. Central to this theory is the concept of parental investment, defined as any parental expenditure in an individual offspring that increases the offspring’s chances of survival and reproductive success, while simultaneously reducing the parent’s ability to invest in other existing or potential offspring. This definition underscores the fundamental trade-off between offspring quantity and quality, reflecting decisions about how resources are allocated among individual offspring.

Trivers’s parental investment theory offers a solid basis for understanding adaptive parental strategies, sexual selection, and variations in reproductive success between sexes. Applying this theoretical framework elucidates many aspects of human reproductive behavior, from mating strategies to investment in offspring, thus providing insights into the evolutionary mechanisms shaping these fundamental biological and social patterns.

Australian biologist David Haig [19] developed the adaptive theory of prenatal conflict, which arises from divergent evolutionary interests between mother and fetus during pregnancy. This conflict is rooted in the fact that the fetus seeks to maximize maternal investment to enhance its own chances of survival and future reproductive success, while the mother must balance current investment in a particular fetus against retaining resources for potential future offspring. The prenatal conflict can manifest physiologically through conditions such as gestational hypertension or gestational diabetes, reflecting the tension between fetal demands and maternal investment capacities.

In extreme cases, intense conflicts over parental investment may lead to pregnancy termination. Such termination may be spontaneous, such as miscarriage, or deliberate, including induced abortions as evolutionarily adaptive maternal responses to adverse environmental conditions [19]. Crucially, this conflict does not cease at birth but continues postnatally.

Postnatal conflict in parental investment is reflected in various maternal decisions regarding the continuation or cessation of investment in offspring. Following childbirth, a mother’s initial decision concerns whether to continue investing in the newborn or to cease investment. Decisions to terminate investment may be direct (infanticide) or indirect (neglect, reduced caregiving, or withholding nutrition). Research has shown that these decisions are not pathological but adaptive strategies responding to ultimate factors such as resource availability, social support, maternal status, and health conditions [1,3].

Understanding these conflicts within reproductive strategy contexts allows insights into the interconnectedness of various reproductive outcomes, including stillbirth, infant mortality, and infanticide. Although proximal mechanisms (immediate physiological or psychological causes) of these outcomes may often be unclear at the individual level, it is evident that they represent responses to similar ultimate evolutionary pressures.

For example, mothers experiencing high stress without adequate social support or stable resources may be more prone to spontaneous abortion or infanticide [20]. While legally and socially distinct, both phenomena represent evolutionary forms of parental investment cessation, whereby mothers adaptively regulate resource allocation between current and future offspring.

Similarly, Haig’s prenatal conflict theory provides a framework for understanding miscarriage as a potentially adaptive maternal strategy, where maternal physiology responds automatically to high stress, resource scarcity, or poor socioeconomic conditions to preserve maternal reproductive potential. Thus, stillbirth and infant mortality form part of a continuum of adaptive responses with a shared evolutionary objective—optimizing parental reproductive success.

Evolutionarily, infanticide, stillbirth, and infant mortality share common ultimate causes, implying that interventions targeting these underlying socioeconomic factors can effectively reduce their incidence. Strategies aimed at improving resource availability, socioeconomic conditions, social support, and reducing environmental stressors constitute effective interventions for improving outcomes, even without complete clarity on individual-level proximal mechanisms [1,20].

Strategic parental investment decisions, whether prenatal or postnatal, fundamentally shape reproductive outcomes. Although proximal mechanisms are not always clearly identifiable, ultimate factors such as resource availability, stress, and social support crucially shape adaptive parental strategies. Focusing on these ultimate factors can enhance our understanding of the biological and evolutionary foundations of human parental behavior and inform effective interventions to reduce adverse reproductive outcomes, including stillbirth, infant mortality, and infanticide.

Infanticide is a complex phenomenon influenced by evolutionary, social, and cultural factors. Evolutionary theory suggests that the ultimate forces shaping population-level rates of stillbirth and infant mortality also influence rates of infanticide, although the proximal mechanisms driving individual cases may remain unclear. Importantly, improving ultimate conditions (such as socioeconomic stability and resource availability) positively impacts individual outcomes, even without a complete understanding of proximal causes.

Phenomena such as infanticide, stillbirth, and infant death represent significant research areas in social sciences, especially within social medicine and psychiatry. Although often studied separately, these events may share common ultimate factors, even when their proximal mechanisms remain unknown or unclear at the individual level. Pioneering studies conducted in Great Britain have substantially contributed to understanding this complex phenomenon.

Marks and Kumar [5,21,22] conducted studies revealing that children under one year in England and Wales are four times more likely to become homicide victims compared to older children and the general population. These figures, likely underestimated due to unreported or misclassified cases, underscore significant challenges in accurately identifying infanticide.

Demographic and psychological characteristics of mothers committing infanticide were further explored, highlighting distinctive profiles, motivations, and social conditions associated with these tragic outcomes [23]. Overall, addressing ultimate socioeconomic and environmental conditions remains essential for reducing incidences of parental investment termination across human societies.

A secondary analysis of d’Orban’s data conducted by Marks and Kumar [22] further demonstrated that mental illness is more prevalent among mothers who commit filicide involving older children (over six months of age), whereas mothers who kill younger children are frequently characterized as abusive without necessarily showing signs of mental illness. This indicates an important distinction between infanticides involving younger versus older children, where the psychiatric background of mothers varies according to the child’s age.

Parental investment cessation is a demographic event of significant importance due to its direct impact on reproductive success, offspring survival, and overall population dynamics. Demographic and evolutionary approaches complement each other [24]: while evolutionary demography analyzes population changes through selective pressures, formal demography provides tools for measuring and interpreting specific events within populations. Different patterns of investment cessation (such as infanticide, neglect, or reduced care) have diverse proximate causes (social, economic, environmental), yet all are deeply rooted in a common evolutionary logic aimed at optimizing reproductive success within specific ecological contexts.

According to life history theory, organisms have limited energetic resources that must be allocated among growth, body maintenance, reproduction, and offspring care [25,26]. When parents face challenges that drastically alter the balance between available resources and potential reproductive success, an adaptive shift in investment can occur, occasionally leading to the complete termination of investment in offspring who have poor chances of survival or reproductive success.

On a proximal level, cessation of investment often arises from specific environmental or socioeconomic conditions, such as extreme hunger, child illness, or high costs associated with additional offspring. However, the ultimate causes of these decisions lie deeper—parents implicitly face evolutionary pressures to optimize their fitness, sometimes requiring resource allocation to offspring with the best prospects. Variations in the forms of investment cessation reflect differences in conditions that determine the optimal strategy available at a particular moment, rather than differences in evolutionary objectives [11,27].

While all forms of parental investment termination, including spontaneous abortion, neglect, and infanticide, share an overarching linkage to socioeconomic stress, it is critical to recognize the differences in proximate mechanisms and intentionality. Spontaneous abortion, often a physiological response to acute stress or resource scarcity, contrasts significantly in mechanism and intentionality with fatal neglect or active infanticide. Thus, these phenomena should be viewed as distinct yet interconnected outcomes along a gradient of parental adaptive responses to socioeconomic stress and resource constraints.

In humans, patterns of parental investment can range from outright infanticide in extreme circumstances, through neglect or reduced care in moderately adverse conditions, to increased investment in offspring quality in favorable environments with lower mortality and better resource availability [28,29]. Despite differing immediate responses, all these strategies can be interpreted as optimal adaptations to specific ecological conditions, making them different manifestations of the same ultimate goal—fitness maximization [25].

From an evolutionary perspective, parental investment strategies have developed to maximize reproductive success under changing ecological, social, and demographic conditions. However, within these strategies, when conditions become too harsh or uncertain, selective cessation of investment can occur, whether through spontaneous abortion, infanticide, or fatal neglect [11].

Human behavioral ecology assumes human behavior to be adaptive, aligned with ultimate causes rooted in natural selection, consciously ignoring intermediate mechanisms (e.g., psychological, hormonal, cultural) responsible for specific behaviors [9]. This methodological approach, known as the “phenotypic gambit”, enables researchers to directly link behaviors and environmental factors [30].

Parental investment cessation (from abstinence, contraception, abortion, and ultimately infanticide) is viewed in behavioral ecology as an adaptive measure protecting the phenotype’s (parent’s) interests and future reproductive success [1].

Although infanticide in most animal species is predominantly associated with unrelated males, in humans, mothers are the most frequent perpetrators, as demonstrated by numerous ethnographic and historical studies [2,31]. This prevalence is due to the demanding and prolonged nature of human parental investment, leaving mothers with limited alternatives under poor social and economic conditions, ultimately resulting in investment cessation.

In contemporary societies, demographic transitions have significantly reduced mortality and birth rates. The availability of contraception, abortion, and institutions caring for unwanted children has dramatically reduced infanticide incidence. Nevertheless, infrequent but ongoing cases of infant murder indicate that, under extremely adverse socioeconomic or psychological conditions, mothers (and/or fathers) may still resort to radical investment cessation [5].

Simultaneously, the “low fertility paradox” is evident: the more socioeconomically developed a society, the more consciously parents control the number of children, prioritizing “quality” over “quantity” [11]. Such strategies suggest a continuous effect of selective parental investment: when the environment is highly competitive, demanding significant investment (higher education, prolonged financial dependence), parents reduce the number of children. Understanding “ecological demography” [32], which observes natality, mortality, and parental investment patterns as phenotypic responses to environmental conditions, is essential in this context.

In societies that have completed demographic transitions, timely forms of pregnancy termination (e.g., legal abortion) and accessible contraception largely replaced infanticide. However, variations in filicide incidence remain associated with extremely adverse conditions (mental illness, unsupportive environments, extreme poverty). Parents choose to cease investment when further care jeopardizes their future reproductive success or survival.

Within human behavioral ecology, infanticide and other forms of parental investment cessation can be understood as adaptive strategies that enable parents to optimize the allocation of their resources among current and future offspring. These strategies stem from fundamental principles of life history theory, including the following:

Trade-off between current and future reproduction: If conditions are unfavorable, parents may opt to terminate investment in one child to enhance the survival prospects of future offspring.

Allocation of limited resources: When resources are scarce, parents must decide whether to invest in one child at the expense of another or prioritize their own survival to potentially have more offspring in the future [33].

Reproductive value of the child: Children with lower survival prospects (due to weakness, illness, or adverse social circumstances) may be deemed less valuable in terms of parental investment, thereby increasing the likelihood of investment cessation [10].

Sara Hrdy [1] thoroughly examines the reasons behind maternal infanticide in humans, emphasizing that mothers typically resort to infanticide when alternative birth control methods such as contraception or institutional care are unavailable and when mothers are unwilling or unable to fully commit to child-rearing. In human societies, infanticide often occurs with minimal violence [34], distinguishing it from the predominantly violent and typically unrelated-perpetrator infanticide observed in other mammals [3].

Evolutionary human behavioral ecology views infanticide as an extreme form of parental investment cessation within a continuum encompassing a broad spectrum of behaviors, ranging from complete commitment to total withdrawal of parental investment [1]. A mother’s decision to cease investment may depend on numerous factors, including resource availability, support from partners or family, child’s health and potential, number of existing children, and anticipated future reproductive success.

From a biological standpoint, abandonment and infanticide represent the same phenomenon—cessation of parental investment. However, societal and legal perceptions of these two situations differ significantly. Active child murder is legally and morally condemned, whereas passive neglect resulting in death often escapes legal sanction, despite identical ultimate outcomes [35].

Research among Mukogodo and Maasai populations [1] provides additional examples of how parental reproductive strategies adapt to social structures and economic conditions. Although the Mukogodo formally express a preference for sons, economic realities lead to preferential investment in daughters. Such adaptive strategies illustrate how parental investment can be shaped by environmental conditions, resulting in varying child mortality rates by sex.

Human behavioral ecology and demography intersect in studying fertility, mortality, and parental investment. These demographic patterns reflect individual parental decisions regarding reproduction and childcare within environmental and societal constraints. The population-based approach of ecological demography [32] assumes that human reproductive behavior evolved through natural selection and results from phenotype–environment interactions. This research aims to test human behavioral ecology hypotheses by analyzing population-level data.

At any given time, a potential mother must balance her development (somatic effort) and offspring investment (reproductive effort). In modern societies, somatic effort frequently involves education and career development [36]. These investments increase reproduction costs, resulting in delayed childbirth and reduced fertility [37]. Similar mechanisms occur in traditional societies; for example, !Kung mothers in the Kalahari Desert adjust birth intervals based on environmental risks—carrying children longer in predator-rich areas, thereby postponing subsequent conceptions [38].

Increased parental investment leads to reduced child mortality rates, including decreases in stillbirths and infant deaths. In competitive societies, such as modern industrialized countries, investment in offspring increases while fertility rates decline as parents strive to maximize their children’s long-term reproductive success [39]. Conversely, some research suggests that high fertility with lower per-child investment may be evolutionarily advantageous, increasing the number of surviving offspring who can later reproduce [40].

Traditional demography focuses on socioeconomic correlates of demographic transition, namely changes in fertility and mortality in modern societies [41]. In contrast, human behavioral ecologists seek to explain these changes through interactions between reproductive strategies and environmental conditions. In traditional societies, greater wealth often correlates with higher fertility, whereas higher socioeconomic status in contemporary societies typically leads to lower fertility [42].

Ruth Mace [43,44] argues that parents adjust family size and investment level based on wealth inheritance. In societies where resources are inherited by offspring, parents limit family size to optimize each child’s success chances. Consequently, increased investment in fewer children reduces mortality and makes additional children unnecessary.

Hillard Kaplan and Jane Lancaster [11,45] highlight competitiveness in modern market societies. Their model suggests prolonged education and child investment are essential for offspring to attain high socioeconomic status, resulting in delayed childbirth and reduced fertility in subsequent generations due to escalating per-child investment. Research involving 7000 men in Albuquerque, USA indicated that ultimate reproductive success might not directly correlate with socioeconomic status [11]. However, socioeconomic status can become critical for survival and reproductive success during ecological or social stressors such as wars or economic crises.

## 4. Dark Figures

Due to advancements in forensic and public health systems, modern practices strictly distinguish between Sudden Infant Death Syndrome (SIDS) and violent deaths. However, research [4,5] warns that some recorded infant deaths may represent concealed infanticides or fatal neglect. For instance, the Confidential Enquiry into Stillbirths and Deaths in Infancy (CESDI) study revealed that in 6.4% of officially diagnosed SIDS cases, there was significant suspicion of violence as the cause of death, while abuse was identified as a secondary factor in an additional 8.1% of cases [4].

The risk of “diagnostic substitution” often arises when there are no clear forensic signs of violence, such as suffocation with a pillow, mild poisoning, or chronic starvation. Such cases underscore the necessity of a multidisciplinary approach in the autopsy and investigation of sudden infant deaths [4].

The ‘dark figure’ refers to cases of infanticide and child neglect that remain undetected and unreported due to methodological, cognitive, or social biases. ‘Diagnostic substitution’, meanwhile, describes situations wherein deaths from infanticide or neglect are inaccurately classified as natural deaths (such as Sudden Infant Death Syndrome, SIDS), due to forensic limitations or cultural biases against recognizing violence in child deaths. Clearly identifying and addressing these concepts is critical for accurately understanding the prevalence and underlying causes of child mortality, as well as the necessity of improving socioeconomic conditions as preventative measures.

One of the biggest challenges in uncovering infanticide is the inconsistency of forensic and legal standards. Studies indicate significant variability between regions regarding the approach to autopsies and criminal investigations of infant deaths. In many cases, standardized forensic tests are not conducted, making it difficult to conclusively exclude the possibility of violent death [4].

The judicial system also exhibits certain gender and socioeconomic biases in prosecuting infanticide cases. Mothers who commit infanticide often receive lighter sentences or are acquitted due to mental disorders, whereas fathers involved in such crimes are more frequently convicted and punished severely [46]. Furthermore, research indicates that poorer families and mothers from marginalized groups face higher scrutiny and are more likely to be prosecuted, while cases involving families of higher socioeconomic status are less frequently subjected to thorough investigations.

## 5. Ultimate Context and Socioeconomic Status

Although the proximal mechanisms leading to infant mortality often remain unclear—including concealed infanticide masked as stillbirth or Sudden Infant Death Syndrome (SIDS)—scientific research confirms that the same ultimate factors can be traced historically and across diverse cultures.

While economic factors play a substantial role in influencing infant mortality, it is essential to recognize the complexity of these phenomena. Socioeconomic stress interacts with numerous other critical factors, including cultural norms regarding childcare, maternal mental health conditions such as postpartum depression, availability of psychological support, and domestic violence. These additional determinants significantly modulate the impact of socioeconomic status on parental investment decisions and child mortality outcomes, indicating the necessity of multifaceted interventions beyond mere economic measures.

While this review primarily applies human behavioral ecology to interpret parental investment cessation, it acknowledges complementary explanatory models, such as those emphasizing individual psychological conditions or cultural-specific practices, which provide valuable context for understanding variability in parental behavior across populations.

Key risk factors for neonaticide include socioeconomic variables such as lack of social support and financial difficulties [47]. Friedman et al. [46] found that poverty, unemployment, and isolation significantly increase the risk, noting that women who commit filicide are frequently primary caregivers with minimal social resources.

Epidemiologists have demonstrated that a woman’s nutritional status before and during pregnancy is critical for healthy pregnancy outcomes [48]. A low maternal body mass index (BMI) correlates with restricted fetal growth and reduced newborn survival. Furthermore, restricted fetal growth does not directly lead to newborn mortality; however, it indirectly contributes to death, especially through asphyxia and various infections, which together constitute approximately 60% of newborn deaths [49]. It is plausible that refraining from reproductive effort under poor maternal conditions and preventing adverse pregnancy outcomes accounts for these 60% of mortality causes linked to maternal health. However, the remaining 40% of mortality causes are not associated with maternal conditions. Consequently, although a decline in stillbirth rates is observed, no statistically significant correlation with a reduction in the number of live births is identified.

The lower the socioeconomic status (SES), the higher the likelihood of poor health or death [50]. Health access is unequally distributed—social and economic resources significantly influence the chances of a long and healthy life. The heaviest burden of disease is borne by individuals with the lowest education, low income, and physically demanding occupations. Structural policies (e.g., education, tax redistribution, public health measures) can mitigate health disparities between social strata.

A critical finding is that the relationship between SES and health is not binary (rich versus poor) but rather graded [51]. Health improves incrementally with each step upward in the social hierarchy (education, occupation, income). This implies that even individuals from the “lower-middle class” experience poorer health than those from higher social strata, despite neither group being impoverished. The authors confirmed this hypothesis in the context of infant mortality as well.

SES is not merely a background variable—it is a cause. SES influences outcomes through the following:

Physical environment: exposure to toxins, overcrowding.

Social environment: violence, social support, resources.

Psychological factors: stress, sense of control, depression.

Health behaviors: smoking, nutrition, physical activity.

The methodological reliance on the ‘phenotypic gambit’, which emphasizes direct environmental–behavioral linkages, inevitably overlooks crucial mediating factors. We have to acknowledge that maternal mental health issues such as postpartum depression, culturally specific parenting norms, and accessibility to psychological support significantly mediate how socioeconomic status ultimately influences parental investment decisions.

Termination of parental investment can occur at all stages of the reproductive cycle—from pre-conception (contraception) to the postnatal period (infanticide, neglect). Depending on the cause and willingness, it can be intentional or unintentional, spontaneous or induced, directly or indirectly caused. Kim and Saada [6] analyzed the impact of SES on pregnancy outcomes in Western countries, identifying significant differences in perinatal and neonatal mortality rates across different socioeconomic groups both between and within countries. Another study [52] demonstrated that higher SES positively affects pregnancy outcomes, including reduced rates of stillbirth and neonatal mortality.

Research in Spain showed that lower SES increases the risk of stillbirth [7]. Another study highlighted a clear connection between SES and elevated risks of stillbirth and perinatal mortality in poorer communities [8]. A review article [53] underscored that poverty and low SES significantly contribute to adverse pregnancy outcomes, including stillbirth and perinatal mortality.

Lower socioeconomic status impacts pregnancy outcomes through a combination of biological, psychological, social, and environmental factors. Key mechanisms include limited access to healthcare, nutritional deficiencies, increased stress, toxin exposure, unhealthy lifestyles, and poor working conditions. Understanding these factors is crucial for creating public health policies aimed at reducing socioeconomic inequalities and improving pregnancy outcomes. All mechanisms linking SES with pregnancy outcomes (stillbirth, perinatal, and neonatal mortality) can also affect infanticide rates, albeit with some specific differences in their mechanisms of action.

Women with lower SES often have restricted access to prenatal healthcare, potentially leading to delayed detection of pregnancy complications [54]. Hospitals and clinics in poorer areas typically have fewer resources, lower funding, and fewer qualified medical professionals. In many countries, women with lower SES lack access to paid maternity leave, resulting in premature return to work and increased stress. Pregnant women from lower socioeconomic backgrounds frequently receive inadequate medical care and face inferior treatment in hospitals.

Healthcare costs, especially in countries without universal healthcare, can limit visits to doctors and access to necessary medications or treatments. Poor nutrition can lead to deficiencies in iron, folic acid, calcium, and other essential nutrients, increasing the risk of preterm birth, low birth weight, and stillbirth [54]. Women with lower SES are often subjected to increased stress due to financial insecurity, unstable employment, and poor living conditions. Prolonged exposure to elevated cortisol levels can adversely affect fetal development and increase the risk of preterm birth [55].

Mental disorders during pregnancy diminish self-care, raising the risk of complications. Hrgović [56] highlights a consistent inverse relationship between SES and mental health; lower socioeconomic status correlates with higher psychopathology rates. Post-World War II research [57] confirmed that individuals from poorer classes more frequently develop disorders due to increased exposure to stressors and limited access to coping resources. Mental disorders result from structurally embedded inequalities, such as unemployment, poverty, and social marginalization. Inequality in the distribution of material and psychosocial resources leads lower socioeconomic groups to suffer more, not due to inherent weakness, but because they face more harmful social conditions.

Poor living conditions, overcrowding, and inadequate sanitation infrastructure elevate infection risks and other pregnancy complications [58]. Studies indicate that women with lower SES are more frequently victims of domestic violence, resulting in fetal injuries, miscarriage, and low birth weight [59]. Research shows that women from lower socioeconomic groups more commonly smoke during pregnancy, increasing risks of stillbirth and sudden infant death syndrome (SIDS) [60]. Alcohol and drug consumption are associated with severe fetal abnormalities, fetal alcohol syndrome, and increased perinatal mortality [61]. Physically demanding work during pregnancy, particularly in industrial or agricultural sectors, raises the risk of preterm birth and stillbirth [62].

All mentioned mechanisms through which SES affects perinatal and neonatal mortality may also influence infanticide rates, though through different pathways, including postpartum depression, economic insecurity, domestic violence, and substance abuse. While perinatal and neonatal mortality typically stem from medical and biological factors, infanticide frequently arises from social, economic, and psychological pressures. Nonetheless, the connection among these phenomena is clear, as they share the same fundamental structural causes.

## 6. Example: The City of Zagreb and Surrounding Municipalities

The theoretical assumptions presented in earlier sections robustly draw from a substantial body of previous research, clearly demonstrating the relationship between socioeconomic status (SES) and various forms of child mortality. Numerous studies have established how lower-SES environments enhance mortality through proximal mechanisms, including restricted access to healthcare, inadequate nutrition, substandard living conditions, limited educational attainment, and heightened stress levels. This existing literature provides a strong theoretical foundation for expecting that SES significantly shapes parental investment decisions and the subsequent outcomes related to child survival and health.

Particularly notable within these correlated forms of mortality is infanticide, a phenomenon inherently challenging to measure accurately due to technical limitations and cognitive biases—frequently leading to misclassification as natural deaths, such as Sudden Infant Death Syndrome (SIDS) or unexplained infant fatalities. As discussed in the earlier sections, including the compelling narrative of “dark figures”, the true incidence of infanticide is likely considerably underestimated in official statistics. This analysis thus emphasizes the critical insight that all forms of child mortality, including concealed infanticide, share common underlying proximal mechanisms driven by socioeconomic conditions.

This illustrative case study draws upon demographic data from Zagreb and surrounding municipalities between 1968 and 2001, sourced from registers at the Institute of Forensic Medicine and Criminology and annual Croatian Bureau of Statistics reports [63]. The statistical correlations presented should be interpreted as indicative rather than exhaustive empirical evidence. The empirical analysis conducted in this study contributes uniquely by demonstrating that, irrespective of how total birth rates fluctuate, distinct forms of child mortality—such as stillbirths, natural-cause infant deaths, accidental deaths, and infanticides—consistently move in parallel and are statistically significantly correlated. Further research, incorporating comprehensive individual-level data, would strengthen causal inferences.

In accordance with the expectations derived from demographic transition theory, the study clearly identified a significant positive correlation between the number of live births and overall child mortality across all ages and causes, including stillbirths, infant deaths from natural causes, accidental deaths, and infanticides. Statistically, a strong correlation was demonstrated (r = 0.784; *p* < 0.001), indicating that a decrease in the number of live births is accompanied by a corresponding decrease in child mortality rates.

When examining specific forms of child mortality, the correlation between the total child mortality rate and the incidence of infanticide is particularly highlighted (Figure 1). A strong correlation is observed between total child mortality and stillbirths (r = 0.814; *p* < 0.001), infant mortality due to natural causes (r = 0.719; *p* < 0.001), and a somewhat weaker yet significant correlation with accidental deaths (r = 0.333; *p* < 0.05). These findings suggest that infanticide is not an isolated phenomenon but part of a broader spectrum of consequences stemming from the socioeconomic conditions in which families live.

As the reduction in infant mortality reflects a parental tendency to invest more heavily in existing children and to abstain from reproductive efforts until reaching a desired level of somatic effort, this increased somatic effort (particularly by the mother) leads to less-exhausting pregnancies and thus reduces the likelihood of stillbirth. The relationship between poor maternal health and adverse birth outcomes has long been recognized in pediatrics [64]. Additionally, increased caution regarding conception might mediate a reduction in future cases of infanticide. If parental investment is discontinued at the point where it is lowest—that is, by abstaining from conception—this reduces future infanticide rates. Conversely, if reproductive effort and subsequent parental investment are focused more intensively on the child already born (while refraining from additional conceptions), the likelihood of child mortality and infanticide decreases.

Under the assumption that all ultimate causes of child mortality derive from socioeconomic status, regardless of immediate or proximal causes, these results further emphasize the need for an in-depth understanding of the social and economic factors that shape living conditions and reproductive decisions of individuals and families. In line with this assumption, socioeconomic status can be considered the fundamental cause underlying all analyzed forms of mortality. Families of lower socioeconomic status are more frequently exposed to stressors such as financial insecurity, limited access to healthcare, educational barriers, and increased psychosocial pressures, which consequently elevate the risks of various forms of child mortality.

In cases of stillbirth and natural-cause infant mortality, socioeconomic factors such as access to adequate prenatal care, nutritional quality of the mother’s diet, living conditions, and health awareness play critical roles. Poverty and social isolation hinder access to resources capable of preventing or mitigating risks associated with these forms of mortality, as evidenced by the strong correlation observed in the research.

Regarding accidental deaths, socioeconomic status may influence outcomes through insufficient protection of children in their physical environment, lower awareness of safety and prevention measures, and inadequate resources for creating a secure environment. Families with lower socioeconomic status more commonly reside in overcrowded, hazardous, and inadequate housing conditions, potentially increasing the risk of accidents and fatal outcomes for children.

Infanticide, as an extreme form of discontinuing parental investment, can be viewed as a direct consequence of severe socioeconomic and social deprivation. Such extreme situations may result from desperation and pressure experienced by parents facing limited resources, lack of societal support, and extreme stigmatization, particularly in cases of unwanted pregnancies or children born out of wedlock. Infanticide, in this context, is not merely an individual moral failure but rather a complex social phenomenon driven by extreme socioeconomic pressures.

## 7. Conclusions: Live Children in a Healthy Society

This review’s main contribution is the systematic use of a human behavioral ecology framework to interpret diverse adverse reproductive outcomes—such as stillbirths, infant mortality, and infanticide—as parts of a unified continuum. This integrative approach offers more consistent theoretical insights and supports the development of more precisely targeted public health strategies.

We can conclude that socioeconomic status is a fundamental variable shaping parental reproductive decisions and the risks to which children are exposed from conception onward. This highlights the need for a holistic approach to policy and interventions aimed at reducing child mortality, which must address social inequalities and economic marginalization of families.

This paper further underscores the importance of public health strategies focused on improving living conditions, education, access to healthcare services, and social support as essential steps toward reducing child mortality from all causes. Targeted social programs addressing the socioeconomic determinants of child mortality are necessary to decrease rates of child mortality, including extreme outcomes such as infanticide.

Ultimately, the findings, interpreted through this assumption, clearly indicate that socioeconomic status is a decisive factor influencing demographic trends and child mortality patterns. Therefore, future research should continue to investigate precisely these socioeconomic dimensions of child mortality, aiming to develop more effective social interventions and policies addressing these underlying causes.

## Figures and Tables

**Figure 1 ijerph-22-00944-f001:**
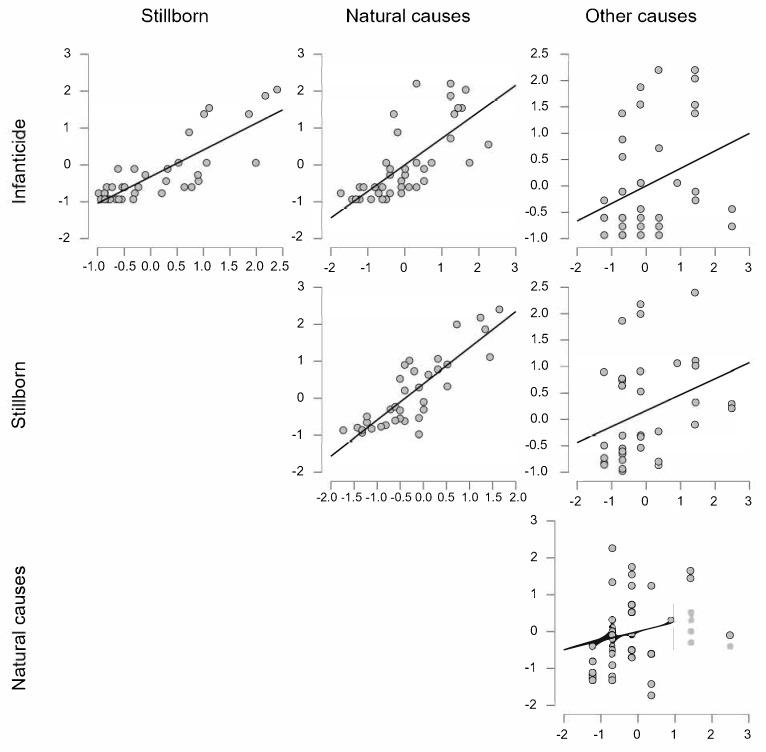
Correlations between infanticide rates and other forms of infant mortality in Zagreb and surrounding municipalities throughout a period of 45 years (1961–2006). Each graph represents a scatterplot showing the Pearson correlation between the incidence of various forms of infant mortality (infanticide, natural causes, stillbirths, and other causes) across years. The incidence rates were previously z-transformed to normalize their distributions. The figure serves as an illustrative example in support of the general hypothesis rather than stand-alone empirical evidence.

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
