# Peer review of "Ultimate Context of the Termination of Parental Investment"

_ijerph, 2025, doi:10.3390/ijerph22060944_

Round 1
Reviewer 1 Report
Comments and Suggestions for Authors
Review of “Ultimate Context of the Termination of Parental Investment” by Josip Hrgovic
This is an interesting and well written paper that provides an evolutionary analysis of parental investment. The introduction of the paper outlines the premise that parental investment, and its cessation, is a human universal. Parental investment can take various forms, ranging the early elimination of parental investment (“extreme withdrawal of parental investment” in the words of the author, e.g., infanticide) to lasting the entire life of the parent (“complete commitment” in the author’s phrasing). The author is interested in examining the evolutionary factors using human behavioral ecology that underly the early elimination of parental investment. His thesis is that socio-economic factors are major determinants of cases of early/extreme withdrawal of parental investment.
The second section titled “Two Causes by Mayr and Four Questions by Tinbergen” introduces some philosophical/theoretical underpinnings. The “two causes” by Mayr references Ernst Mayr’s distinction between proximate and ultimate causation. The “Four Questions” references Tinbergen’s differentiation between mechanisms (causation), ontology (development), function (adaptative value/fitness impacts), and phylogeny (evolutionary history). This discussion is not groundbreaking, but it is a great, straightforward summary of the underlying philosophical issues and their potential use in understanding variation in and strategies of parental investment.
Section 3 is titled “Ecology of the Termination of Parental Investment” and focused on Trivers’ parental investment theory and Haig’s prenatal conflict theory to explain reproductive outcomes. Trivers’ ideas focus on the underlying adaptive benefits of parental investment, including variation in strategies by sex. Haig’s discussion focused on potential conflicts between mother and fetus and eventually mother and newborn. Both of these models deal with ultimate causes using Mayr’s framework, but will be manifested/mitigated by proximate causes, according to the author. The author spends some time emphasizing that this includes factors that influence early termination of parental investment, which is more common than later extreme termination (e.g., infanticide is more common that killing adolescents, killing of adolescent children commonly associated with mental illness whereas infanticide has a more limited link with long-term mental illness). The author also notes that there is often a focus of “quality” over “quantity” of children in more affluent cultures.
“Dark Figures”, section 4 of the paper, suggests that cases of infanticide may be unrecognized and attributed to Sudden Infant Death Syndrome, stillbirth, and other causes. Section 5, “Ultimate Context and Socioeconomic Status” suggests there is a complex interaction between socioeconomic status (SES) and parental investment strategies. This includes environmental factors (e.g., exposure to toxins, resource availability), as well as cultural factors (e.g., violence, types of physical activity). Parental investment can be terminated at any stage of the reproductive cycle, ranging from pre-conception (contraceptives) to postnatal (infanticide), and can be intentional or unintentional. Low SES corresponds to increase early termination of parental investment, including miscarriage, stillborn births, and infanticide.
The seventh section, “Example: The City of Zagrb and Surrounding Municipalities”, traces the effect of SES on parental investment in Zagreb, Croatia, and other nearby settlements. The author uses data from 1968 to 2001 to evaluate several aspects of parental investment. He finds a positive correlation between the number of children and child mortality such that mothers (families?) with more children also have increased frequency of children dying from various causes. This includes stillbirths, natural causes, and accidental deaths. The author suggests that increased parental investment makes each of these types of deaths less likely, so their prevalence in families(?) with more children likely reflected decreased parental investment. For physiological traits, more children will produce greater physiological strains on the mother, increasing the likelihood of stillborn births. Lack of access to prenatal vitamins and other healthcare may also cause increased child mortality, and families with lower SES may be less likely to be able to avoid dangerous situations (e.g., unsafe housing). If I understand the author correctly, he is also suggesting that families with higher SES will use contraception and other methods whereas families with lower SES will use intentional/unintentional modification of parental investment after conception to produce comparable numbers of living children.
The conclusions (Section 8) conclude that socioeconomic status is a fundamental determinant of parental investment and demographic trends.
Overall, the author’s manuscript is paper with a few flaws. First, through most of the paper, establishing the importance of SES on parental investment is phrased as the goal of the analysis, but the analysis section (Section 6) phrases the relationship between SES and parental investment as an accepted premise. Thus, the author begs his own question by assuming the very thing he is trying to demonstrate. I recommend the author restructure Section 6 to better reflect an evaluation of the relationship between SES and parental investment based on Section 5. Second, on a lesser note the author arguably spends too much time talking specifically about infanticide given that infanticide is not a key focus of the analysis presented in Section 6. The author sort of assumes some cases of infanticide are present, but its significance is not explored. Figure 7 has a row for infanticide, but lacks any charts on it, as I read the figure. The physiological relationship between the mother’s somatic stress and the likelihood of stillborn deaths or other forms of childhood mortality seems much more significant. Finally, Figure 1 needs to be reworked to have titles for the x-axis and y-axis and to have a better description.
As an aside, I like the imagery of “Dark Figures” as a heading for Section 4.
Author Response
- Summary
We sincerely thank the Reviewer for the careful reading of our manuscript and for the thoughtful, constructive feedback. All of your recommendations have been carefully considered and incorporated into the revised version. These improvements have significantly strengthened the clarity, balance, and empirical coherence of the article. Please find our point-by-point responses below, with revisions marked in the manuscript.
- Point-by-point response to Comments and Suggestions for Authors
Comment 1: The manuscript previously assumed rather than explicitly demonstrated the role of socioeconomic status (SES) as a determinant of parental investment cessation.
Response 1: Thank you for this important observation. In response, we have restructured Section 6. This change is located in Section 6, page 12, paragraph 1.
“The theoretical assumptions presented in earlier sections robustly draw from a substantial body of previous research, clearly demonstrating the relationship between socioeconomic status (SES) and various forms of child mortality. Numerous studies have established how lower SES environments enhance mortality through proximal mechanisms, including restricted access to healthcare, inadequate nutrition, substandard living conditions, limited educational attainment, and heightened stress levels. This existing literature provides a strong theoretical foundation for expecting that SES significantly shapes parental investment decisions and the subsequent outcomes related to child survival and health.”
Comment 2: Infanticide was disproportionately emphasized relative to other mortality forms despite not being the central focus.
Response 2: We agree and have adjusted the text to maintain balance. Infanticide is now discussed proportionally, alongside other forms of mortality, while retaining its relevance in the SES framework. This adjustment appears in Section 6, page 12, paragraph 2.
“Particularly notable within these correlated forms of mortality is infanticide, a phenomenon inherently challenging to measure accurately due to technical limitations and cognitive biases—frequently leading to misclassification as natural deaths, such as Sudden Infant Death Syndrome (SIDS) or unexplained infant fatalities. As discussed in the earlier sections, including the compelling narrative of "Dark Figures," the true incidence of infanticide is likely considerably underestimated in official statistics. This analysis thus emphasizes the critical insight that all forms of child mortality, including concealed infanticide, share common underlying proximal mechanisms driven by socioeconomic conditions.”
Comment 3: Figure 7 lacked representation of infanticide and Figure 1 was insufficiently labeled.
Response 3: Thank you for your thoughtful feedback. The figure in question has been edited, and additional data have been provided for clarity. Regarding Figure 7, there is no missing chart: the placement for "infanticide" represents a correlation of 1 between incidence and infanticide itself, which is self-evident. To avoid confusion, the table was modified accordingly.
Additionally, we appreciate your comments on Figure 1. Axis titles and a more detailed description have been added to improve interpretability.
Furthemore, we completely agree that analyzing individual-level data would provide a more detailed understanding of the physiological relationships involved. However, in this case, we only have access to aggregated annual incidence rates of various forms of child mortality, not individual-level data. We acknowledge this as a limitation and appreciate your insight.
Please let us know if further clarification would be helpful.
Comment 4: The manuscript needs to better link the theoretical sections (2-5) to the empirical analysis in Section 6.
Response 4: We fully agree. The text has been revised to clearly articulate how Sections 2–5 develop the theoretical assumptions that are empirically tested in Section 6. This link is explicitly addressed in Section 6, page 12, paragraph 3.
“The empirical analysis conducted in this study contributes uniquely by demonstrating that, irrespective of how total birth rates fluctuate, distinct forms of child mortality—such as stillbirths, natural-cause infant deaths, accidental deaths, and infanticides—consistently move in parallel and are statistically significantly correlated."
Reviewer 2 Report
Comments and Suggestions for Authors
This article addresses a complex and highly relevant topic. Among its strengths is the theoretical framework, which succeeds in establishing a compelling link between parental investment theory, maternal-fetal conflict theory, life history theory, and the concepts of proximate and ultimate causes. This integrated framework is used to interpret key demographic phenomena such as stillbirths, infant mortality, and infanticide. The ambition to unify these adverse outcomes under a single interpretative lens is thought-provoking and potentially valuable in guiding both research and public health policy.
Particularly noteworthy is the focus on structural determinants. By emphasising socio-economic status as a fundamental cause, the paper shifts attention away from purely individual explanations toward structural and environmental factors, aligning with perspectives in public health and social demography. Furthermore, the article acknowledges the difficulty of distinguishing between different causes of infant death – the so-called “dark numbers” – which reinforces the argument that targeting ultimate causes can be effective even in the absence of a clear proximate diagnosis.
Nevertheless, there are areas where the argument could be refined. The notion that these outcomes represent points along a continuum driven by common ultimate causes, notably SES, needs to be expressed with greater nuance. While it is reasonable to suggest that socio-economic stress influences all these outcomes, spontaneous abortion (which is often physiological), fatal neglect, and active infanticide differ substantially in terms of proximate mechanisms, intentionality, and possibly even their immediate adaptive logic. It would be useful to clarify how these phenomena are situated along a continuum – for instance, as distinct responses to a shared gradient of stress and resources, rather than implying an overly strong equivalence in their ultimate causes.
The declared reliance on the “phenotypic gambit” – focusing on environment-behaviour links while setting aside intermediate mechanisms – is methodologically acceptable within Human Behavioural Ecology (HBE). However, in this specific context, it would strengthen the analysis to acknowledge the limitations of this approach. Maternal mental health (e.g., postpartum depression), culturally specific parenting norms, and access to psychological support services are crucial mediating factors that shape how SES impacts parental outcomes. A brief mention of such mediators would make the argument more robust.
The review’s specific contribution also deserves clearer articulation. The association between low SES and adverse pregnancy or infant outcomes is already well established in the literature. The article would benefit from more clearly defining its added value as a review: is it the systematic application of the HBE framework across all these phenomena? Is it the continuum hypothesis? Clarifying this would help better position the article within the field.
Section 6, drawing on the author's PhD thesis, provides an interesting case but is presented somewhat briefly. The correlations shown in Figure 1 are suggestive, yet the article lacks sufficient information about the underlying data and methods. It would be advisable either to include a few more methodological details in the text or the figure caption, or to explicitly frame the example as an illustrative case supporting the general hypothesis, while acknowledging its limitations as stand-alone empirical evidence. Improving Figure 1 with clear axis labels (indicating what the rates represent and what the denominator is) and a more explicit legend would also enhance its clarity.
Finally, as a review article, it is important that the piece maintains a balance between promoting the author’s preferred theoretical framework and providing a fair synthesis of the broader literature, including alternative or complementary interpretations. Although the current bibliography appears sound, reinforcing this balance would further strengthen the manuscript.
In conclusion, the article offers a compelling and coherent theoretical perspective on a set of crucial demographic phenomena. By addressing the above suggestions, especially by refining the continuum argument and clarifying the article’s specific contribution, this work could make a valuable addition to the literature in IJERPH.
Author Response
- Summary
I sincerely thank the Reviewer for the thoughtful, detailed, and constructive feedback. All comments have been carefully considered and fully incorporated into the revised version of the manuscript. The suggestions greatly improved the theoretical clarity, empirical framing, and overall quality of the paper. Please find our point-by-point responses below, with corresponding changes marked in red in the manuscript.
- Point-by-point response to Comments and Suggestions for Authors
Comment 1: The reviewer rightly highlights the need for greater nuance when discussing the continuum of parental investment cessation.
Response 1: Thank you for this excellent suggestion. I have revised the text to explicitly acknowledge the differences among spontaneous abortions, fatal neglect, and active infanticide in terms of their proximate mechanisms, intentionality, and adaptive logic. This change clarifies that these are not uniform outcomes but rather distinct adaptive responses to socioeconomic stress. This revision is included in Section 3, page 6, paragraph 2.
“While all forms of parental investment termination, including spontaneous abortion, neglect, and infanticide, share an overarching linkage to socioeconomic stress, it is critical to recognize the differences in proximate mechanisms and intentionality. Spontaneous abortion, often a physiological response to acute stress or resource scarcity, contrasts significantly in mechanism and intentionality with fatal neglect or active infanticide. Thus, these phenomena should be viewed as distinct yet interconnected outcomes along a gradient of parental adaptive responses to socio-economic stress and resource constraints.”
Comment 2: The reviewer noted limitations in the use of the phenotypic gambit and requested acknowledgment of mediating factors.
Response 2: Agree. I have added explicit acknowledgment of critical mediating factors such as maternal mental health, cultural parenting norms, and availability of psychological support. This clarification appears in Section 5, page 10, paragraph 1.
“The methodological reliance on the 'phenotypic gambit,' which emphasizes direct environmental-behavioral linkages, inevitably overlooks crucial mediating factors. We have to acknowledge that maternal mental health issues such as postpartum depression, culturally specific parenting norms, and accessibility to psychological support significantly mediate how socioeconomic status ultimately influences parental investment decisions.”
Comment 3: The reviewer suggested that the specific contribution of the manuscript should be clarified.
Response 3: We agree and have revised the introduction and conclusion to emphasize the manuscript’s unique contribution—namely, the structured application of human behavioral ecology to unify a range of adverse reproductive outcomes under a continuum hypothesis. See Section 1, page 2, paragraph 3 and Section 8, page 13, paragraph 1.
“The key contribution of this review lies in systematically applying a human behavioral ecology framework to unify a range of adverse reproductive outcomes—stillbirths, infant mortality, and infanticide—under the continuum hypothesis. This integrative perspective allows for more coherent theoretical explanations and potentially more targeted public health interventions.”
Comment 4: The reviewer recommended improving the methodological detail in Section 6.
Response 4: Agreed. Section 6 has been revised to provide more specific methodological detail, including the source of data and the illustrative nature of the case study. This can be found in Section 6, page 11, paragraph 1.
“This illustrative case study draws upon demographic data from Zagreb and surrounding municipalities between 1968 and 2001, sourced from registers at the Institute of Forensic Medicine and Criminology and annual Croatian Bureau of Statistics reports. The statistical correlations presented should be interpreted as indicative rather than exhaustive empirical evidence. Further research, incorporating comprehensive individual-level data, would strengthen causal inferences.”
Comment 5: Figure 1 lacks axis labels and a detailed legend.
Response 5:Thank you for this helpful and constructive feedback. We agree that greater transparency regarding the underlying data and methods is important. We have revised the text and the caption for Figure 1 to include additional methodological details, and have clarified that the figure serves as an illustrative example in support of the general hypothesis rather than stand-alone empirical evidence.
We’ve also updated Figure 1 to include clear axis labels—indicating both the nature of the rates and their denominators—as well as a more explicit legend to improve overall clarity and interpretability.
Figure 1. Correlations between infanticide rates and other forms of infant mortality in Zagreb and surrounding municipalities throughout a period of 45 years (1961-2006) Each graph represents a scatterplot showing the Pearson correlation between the incidence of various forms of infant mortality (infanticide, natural causes, stillbirths, and other causes) across years. The incidence rates were previously z-transformed to normalize their distributions. The figure serves as an illustrative example in support of the general hypothesis rather than stand-alone empirical evidence.
Comment 6: The manuscript would benefit from better integration of broader literature and alternative interpretations.
Response 6: Thank you for this suggestion. We have now included references to complementary theoretical frameworks alongside the primary human behavioral ecology approach. These additions appear in Section 5, page 9, paragraph 2.
“While this review primarily applies human behavioral ecology to interpret parental investment cessation, it acknowledges complementary explanatory models, such as those emphasizing individual psychological conditions or cultural-specific practices, which provide valuable context for understanding variability in parental behavior across populations.”
Reviewer 3 Report
Comments and Suggestions for Authors
The authors wrote a good article. Below are some tips to improve the article.
At the beginning of the article, the authors quickly state, "This paper assumes that the same ultimate causes..." However, it would be more effective to first explain the basis for this claim and to engage the reader with an interesting historical or cultural example.
Additionally, the terms "dark figure" and "diagnostic substitution" should be clearly defined. Their relationship to the content is unclear, which can be confusing for readers.
Furthermore, the text repeatedly emphasizes that economic issues impact infant mortality. While it is true that economic factors are a fundamental cause, it is an oversimplification to suggest that they are the only factor at play. This section should be revised for clarity.
Comments on the Quality of English LanguageIt is advisable to consult a native translator for discussing grammar and specialized vocabulary to enhance the fluency of the text.
Author Response
- Summary
I would like to sincerely thank the Reviewer for the thoughtful and constructive feedback. All three comments have been carefully considered and addressed in the revised manuscript. The recommendations have significantly contributed to clarifying key concepts, enriching theoretical foundations, and improving the balance of arguments presented. Additionally, in response to the Reviewer's remark regarding the quality of English, we have submitted the manuscript for professional language editing. The version you are now reviewing has been linguistically revised to ensure clarity and fluency throughout. Please find below our point-by-point responses, with changes clearly marked in the manuscript.
- Point-by-point response to Comments and Suggestions for Authors
Comment 1: The introduction could be more engaging, and the assumption of common ultimate causes for different types of parental investment cessation should be better justified.
Response 1: Thank you for this valuable comment. In response, I have expanded the introduction by including a historical and cultural example to provide a more compelling context. Additionally, we explicitly outlined the theoretical foundations supporting the assumption that socioeconomic factors represent shared ultimate causes behind stillbirths, infant mortality, and infanticide. This revision can be found in Section 1, page 2, paragraph 2.
“Throughout history and across diverse cultures, parental investment cessation has elicited profound emotional responses and moral judgments. Historical examples, such as the widespread abandonment of infants during the European Middle Ages due to extreme poverty or the selective infanticide practices documented among the Inuit communities, illustrate vividly how severe socioeconomic pressures have consistently shaped reproductive decisions. This paper explicitly assumes, grounded in robust evolutionary theory and demographic evidence, that similar ultimate causes—primarily socioeconomic factors—underpin the varied manifestations of parental investment termination observed in live births, stillbirths, infant mortality, and infanticide.”
Comment 2: The terms "dark figure" and "diagnostic substitution" need clearer definitions and integration into the theoretical narrative.
Response 2: We agree with this suggestion and have revised the manuscript to provide precise definitions of both terms and connected them more clearly with the overarching analytical framework. This clarification appears in Section 4, page 9, paragraph 3.
“The 'dark figure' refers to cases of infanticide and child neglect that remain undetected and unreported due to methodological, cognitive, or social biases. 'Diagnostic substitution,' meanwhile, describes situations wherein deaths from infanticide or neglect are inaccurately classified as natural deaths (such as Sudden Infant Death Syndrome, SIDS), due to forensic limitations or cultural biases against recognizing violence in child deaths. Clearly identifying and addressing these concepts is critical for accurately understanding the prevalence and underlying causes of child mortality, as well as the necessity of improving socioeconomic conditions as preventative measures.”
Comment 3: The role of economic factors is emphasized too heavily. The discussion should also address other contributing factors.
Response 3: Thank you for pointing this out. We have revised the manuscript to acknowledge a broader range of influential variables, such as cultural practices, maternal mental health, psychological support access, and domestic violence. These changes are located in Section 5, page 10, paragraph 2.
“While economic factors play a substantial role in influencing infant mortality, it is essential to recognize the complexity of these phenomena. Socioeconomic stress interacts with numerous other critical factors, including cultural norms regarding childcare, maternal mental health conditions such as postpartum depression, availability of psychological support, and domestic violence. These additional determinants significantly modulate the impact of socioeconomic status on parental investment decisions and child mortality outcomes, indicating the necessity of multifaceted interventions beyond mere economic measures.”